# Antiviral Activity of Luteolin against Pseudorabies Virus In Vitro and In Vivo

**DOI:** 10.3390/ani13040761

**Published:** 2023-02-20

**Authors:** Xiaoyu Men, Su Li, Xiaojing Cai, Lian Fu, Yi Shao, Yan Zhu

**Affiliations:** 1College of Veterinary Medicine, Northeast Agricultural University, Harbin 150038,China; 2State Key Laboratory of Veterinary Biotechnology, Harbin Veterinary Research Institute, Chinese Academy of Agricultural Sciences, Harbin 150008, China

**Keywords:** luteolin, pseudorabies virus, antiviral activity

## Abstract

**Simple Summary:**

Pseudorabies virus (PRV) can cause acute swine disease leading to economic losses worldwide and is a potential causative agent of viral encephalitis in humans. Although effective vaccines are available, an increasing number of variants have emerged in China, and identifying effective antiviral agents against PRV to prevent latent infection is essential. Luteolin is the main flavonoid component in honeysuckle and is also found in herbs and other plants, such as chamomile tea, perilla leaf, green pepper, and celery. In this study, we assessed the antiviral activity of luteolin against PRV in vitro and in vivo. Luteolin inhibited the virus at the replication stage and decreased the expression of viral mRNA and gB protein. Luteolin reduced the apoptosis of PRV-infected cells, improved the survival rate of mice after lethal challenge, reduced the viral loads in the liver, kidney, heart, lung, and brain, reduced brain lesions, and slowed inflammation and oxidation reactions. These pieces of evidence suggest luteolin has promise as a new alternative antiviral drug for PRV infection.

**Abstract:**

Pseudorabies virus (PRV) can cause acute swine disease leading to economic losses worldwide and is a potential causative agent of viral encephalitis in humans. Although effective vaccines are available, an increasing number of variants have emerged in China, and identifying effective antiviral agents against PRV to prevent latent infection is essential. In this study, we assessed the antiviral activity of luteolin against PRV in vitro and in vivo. Luteolin was found to significantly inhibit PRV at a noncytotoxic concentration (70 μM), with an IC_50_ of 26.24 μM and a selectivity index of 5.64. Luteolin inhibited the virus at the replication stage and decreased the expression of viral mRNA and gB protein. Luteolin reduced the apoptosis of PRV-infected cells, improved the survival rate of mice after lethal challenge, reduced the viral loads in the liver, kidney, heart, lung, and brain, reduced brain lesions, and slowed inflammation and oxidation reactions. Our results showed that luteolin has promise as a new alternative antiviral drug for PRV infection.

## 1. Introduction

Pseudorabies virus (PRV), a member of the swine herpesvirus Alphaherpesvirinae subfamily, is the causative agent of Aujeszky’s disease, which leads to significant economic losses in the global swine industry [1,2]. Pigs are the natural host and reservoir of PRV, and PRV infection is characterized by encephalomyelitis, reproductive failure in adult animals, inflammation of the upper respiratory tract and lungs, and 100% mortality in newborn piglets due to central nervous system disorders [3]. PRV can also infect numerous other mammals, such as ruminants, rodents, and carnivores [2]. Importantly, several cases of human viral encephalitis caused by PRV infection have been reported. Indeed, PRV-positive antibodies have been detected in encephalitis patients and healthy people [4,5]. Thus, the threat of PRV to humans and other animals should not be ignored. Although vaccines are widely used to control PRV, several studies [6,7] have reported PRV infection in vaccinated pig herds in China. These cases suggest that PRV mutants are important causes of morbidity and mortality and that the present vaccines cannot provide complete protection against PRV variants [8]. Thus, new control measures are urgently needed.

Natural compounds have a wide range of applicability in preventing and treating diseases and have become an important source of novel antiviral drugs. Many studies have shown that flavonoids have a wide range of bioactivities against many diseases, such as tumors, inflammation, and viral diseases [9,10]. Luteolin is a member of the flavone family, 3′4′5′7′-tetra- hydroxyflavone [11]. It is the main flavonoid component in honeysuckle and is also found in herbs and other plants, such as chamomile tea, perilla leaf, green pepper, and celery [12,13]. Luteolin has a wide range of pharmacological activities, such as anti-inflammatory [14,15], anticancer [16,17], antioxidant [18,19], and neuroprotective effects [20]. In addition, luteolin has been reported to exhibit antiviral activities against many viruses, including Japanese encephalitis virus (JEV), dengue virus (DENV), chikungunya virus (CHIKV), influenza virus (IAV), herpes simplex virus type 2 (HSV-2), and African swine fever (ASF) [21,22,23,24,25,26]. Although luteolin is a potential antiviral drug, little is known about the antiviral activity of luteolin against PRV. Therefore, the anti-PRV activity of luteolin in vivo and in vitro was systematically evaluated in this study to provide scientific data for developing a new alternative measure for controlling PRV infection.

## 2. Materials and Methods

### 2.1. Cells, Viruses, and Luteolin

PK15 cells were cultured in Dulbecco’s modified Eagle’s medium (DMEM, Gibco Invitrogen, Carlsbad, CA, USA) supplemented with 10% (*v*/*v*) fetal bovine serum (FBS, Gibco Invitrogen, Carlsbad, CA, USA), 100 U/mL penicillin, and 100 mg/mL streptomycin (Gibco Invitrogen, Carlsbad, CA, USA) at 37 °C in 5% CO_2_. PRV (GenBank accession: KJ789182.1) was a gift from the Harbin Veterinary Institute of China. The Reed-Muench method was used to determine the median tissue culture infectious dose (TCID_50_) of PRV. Luteolin (Meilun, Dalian, China) was dissolved in dimethyl sulfoxide (DMSO). The DMSO concentration of the solution did not exceed 1%.

### 2.2. Cytotoxicity Analysis

Cell viability was detected with a Cell Counting Kit 8 (CCK8, Bimake, Houston, TX, USA) assay to evaluate the cytotoxicity of luteolin. Briefly, 10^4^/mL PK15 cells were cultured in 96-well plates (Costar, USA) at 37 °C in an atmosphere of 5% CO_2_. The cell monolayers were treated with luteolin at 0, 52, 70, 87, 105, 140, and 175 μM for 48 h, with six replicates for each concentration. The cell state was observed under an optical microscope, and the cells in each well were added to 10% (*v*/*v*) CCK8 solution and incubated for 1 h at 37 °C to measure the absorbance at a wavelength of 450 nm. Cell viability was expressed as a percentage of the control (no DAE treatment) cell viability. The 50% cytotoxic concentration (CC_50_) of DAE was calculated with GraphPad Prism 8.0 software (GraphPad Software, San Diego, CA, USA).

### 2.3. Antiviral Act Ivity Assay of Luteolin against PRV In Vitro

To evaluate antiviral activity, we investigated the inhibitory effects of luteolin on PRV. PK15 cells were cultured in a 96-well plate under 5% CO_2_ and 37 °C conditions until they grew to a density of 70–80% confluence. PK15 cells were treated with luteolin 1000 μL (0, 11, 17.5, 22, 35, 44, 70 μM) 1 h after virus infection (100 TCID_50_). After 48 h of culture, the medium was removed, and the cells were washed twice with PBS. Then, the cell state was observed under an optical microscope. CCK8 assay was carried out at 37 °C for 0.5–1 h, and the OD value was measured at 450 nm. The CC_50_ and half maximal inhibitory concentration (IC_50_) were determined by GraphPad Prism software version 8 (GraphPad Software Company, La Jolla, CA, USA).

### 2.4. Extraction and Quantification of Viral DNA

#### 2.4.1. Viral Adsorption Stage

Six-well plates were inoculated with PK15 cells at a density of 10^5^ cells/well. After cooling the cells at 4 °C for 30 min, luteolin (0, 35, 44, 70 μM) and virus (100 TCID_50_) premix were added and adsorbed at 4 °C for 1 h, and then the cells were washed to remove unadsorbed viruses, followed by incubation at 37 °C.

#### 2.4.2. Viral Entry Stage

Six-well plates were inoculated with PK15 cells at a density of 10^5^ cells/well. The PK15 cell monolayer was incubated with the virus (100 TCID_50_) at 4 °C for 1 h, the virus solution was removed, and fresh medium containing luteolin (0, 35, 44, 70 μM) was added.

#### 2.4.3. Viral Replication Stage

Six-well plates were inoculated with PK15 cells at a density of 10^5^ cells/well. The PK15 cell monolayer was incubated with the virus (100 TCID_50_) at 37 °C for 1 h, the virus solution was removed, and fresh medium containing luteolin (0, 35, 44, 70 μM) was added.

Infected cells were harvested 48 h after infection. Total DNA was extracted using a viral DNA kit (Beijing Tiangen Biotechnology Co., Beijing, China). PRV DNA copies were analyzed by quantitative polymerase chain reaction (qPCR). Purified plasmids containing the gI gene were used to generate standard curves. Primer sequences and probes are listed in Table 1.

### 2.5. Gene Expression Analysis

The PK15 cell monolayer was infected with PRV (100 TCID_50)_ for 1 h. Then, the supernatant was removed and replaced with fresh medium containing luteolin. The supernatant was released at 6, 12, 24, 48, and 72 h after infection, and the infected cells were collected. Total RNA was extracted using the RNAprep pure Cell Kit (Tiangen Biotech; Beijing, China). A cDNA synthesis kit was used according to the manufacturer’s instructions (Tiangen Biotech; Beijing, China), and RNA was immediately reverse-transcribed into cDNA. β-Actin was used as a housekeeping gene. The primers used for relative qPCR are shown in Table 1.

### 2.6. Molecular Docking and Western Blot Verification of the PRV gB Protein

AutoDock 4.2 6 was used for molecular docking analysis of gB and luteolin. AUTOGRID supporting software was used to calculate the atomic affinity potential energy. We downloaded the chemical structure of luteolin from the Laboratory of Systems Pharmacy (molecular ID: mol000006) and the X-ray structure of gB (PDB no. 6esc) from the PDB protein structure database.

PK15 cells were cultured to 90% confluence in 6-well plates. After the virus and cells were incubated at 37 °C for 1 h, the supernatant was discarded, and medium containing luteolin was added. After 48 h of culture, the cells were washed with PBS 3 times, lysed with protein extraction buffer at 4 °C for 30 min, and centrifuged at 5000× *g* for 10 min, and the supernatant was collected. SDS–PAGE (10%) was used to separate the protein in the supernatant, which was transferred to a PVDF membrane at 300 mA for 100 min. Five percent skimmed milk was used to block the membrane at room temperature for 2 h. The membrane was incubated overnight at 4 °C in primary antibodies, which were diluted with TBST (1:10,000), against PRV gB and β-actin. The PVDF membrane was incubated with anti-mouse or anti-rabbit IgG peroxidase-coupled secondary antibodies at room temperature for 1 h. ECL™ was used to observe the proteins bound to the membrane, and the signals were analyzed using ImageJ software (National Institutes of Health, Bethesda, MD, USA).

### 2.7. Apoptosis Analysis

PK15 cells were incubated with virus for 1 h, supplemented with luteolin-containing medium, cultured for 48 h, digested with trypsin without EDTA, and centrifuged at 1000 rpm for 5 min. The cells were resuspended in PBS, centrifuged at 1000 rpm for 5 min, and then mixed with 500 µL 1 × Mix with buffer. After adding 5 µL annexin V-FITC and 5 µL PI mixture into each tube, the cells were incubated in the dark at room temperature for 10 min. Annexin V-FITC was detected by the FITC detection channel (ex 488 nm; em 530 nm), and PI was caught by the PI detection channel (ex 535 nm; em 615 nm) using flow cytometry.

### 2.8. Antiviral Activity Assay of Luteolin against PRV In Vivo

To evaluate the anti-PRV activity of luteolin in vivo, we used a mouse model of PRV muscle infection. Five-week-old female BALB/c mice (18–20 g) were purchased from Changsheng Biotechnology Co., Ltd. (Harbin, China) and were randomly divided into the following three groups with six mice in each group: normal control group (uninfected and untreated), virus group (infected and untreated), and treatment group (infected and treated with 100 mg/kg/D luteolin) (Table 2).

One hundred microliters of PRV (10^3^ TCID_50_/mL) was used to establish PRV infection in mice by intramuscular injection. Luteolin was injected intraperitoneally 1 h after injection and once a day for three consecutive days. The survival rate, death protection, average survival time, and weight changes were observed daily for 7 d. The mice in each group were euthanized, and the liver, heart, brain, lung, and kidney tissues were collected and cryopreserved at −80 °C. The brain tissues of 2 mice in each group were fixed with 4.0% paraformaldehyde, embedded in paraffin, and stained with hematoxylin-eosin (HE) solution. One hundred milligrams of liver, heart, brain, lung, and kidney tissue from each group was ground evenly, the total DNA of the tissue samples was extracted according to the method described in Section 2.3, and the virus copy number was detected by fluorescence qPCR. Blood was collected from the eyeball vein, left at 37 °C for 30 min, and centrifuged at 10,000× *g* for 10 min. Serum was collected, and IL-4, IFN-γ, and TNF-α levels in serum were determined by ELISA kits (Beijing Chenglin Biotechnology Co. Ltd., Beijing, China).

### 2.9. Statistical Analysis

The results are expressed as the mean and standard deviation (SD). For statistical analysis, Prism 8 (GraphPad Software, La Jola, CA, USA) was used. Significance was determined by one-way analysis of variance (ANOVA) and two-way ANOVA with Dunnett’s multiple-comparison test. All experiments were performed in triplicate. Values of *p* < 0.05 were statistically significant at a 95% confidence interval.

## 3. Results

### 3.1. Cytotoxicity of Luteolin

As shown in Figure 1, when the luteolin concentration exceeded 70 μM, the activity of PK15 cells was significantly inhibited (*p* < 0.0001). The CC_50_ was 148.1 μM ± 1.15, as shown in Table 3. Then, we explored the anti-PRV activity of luteolin at 70 μM and lower concentrations.

### 3.2. Antiviral Activity of Luteolin

We measured the inhibitory effects of luteolin on PRV within 48 h by the CCK8 method. We found that luteolin has a strong inhibitory effect. At a concentration of 70 μM, the highest inhibition rate was 86.39%, and the IC_50_ was 26.24 ± 0.01 (Figure 2).

To determine the effect of luteolin on viral adsorption, invasion, and replication, FQ-PCR was used to detect viral copies in cells after 48 h. Luteolin can inhibit these three stages to varying degrees but mainly affects the viral replication stage, as shown in Figure 3. In the viral replication stage, the PRV copies of the treatment group with the highest luteolin concentration were 10.06 times lower than those of the untreated group, indicating significant inhibition of viral proliferation.

### 3.3. Inhibitory Effect of Luteolin on PRV Gene Expression

To study the effect of luteolin on PRV gene expression, we detected the expression of the immediate early gene, early gene, and other genes required for PRV replication. Figure 4 shows the relative expression levels of the tested genes (IE180, EPO, UL27, UL29, UL9, and UL54). The relative expression of all detected genes in the luteolin treatment group was lower than that in the virus group, but it still showed an upward trend. However, luteolin significantly inhibited this proliferation trend compared with the virus group. Within 72 hpi, 70 μM luteolin significantly inhibited the expression of all tested genes. The results showed that luteolin could inhibit viral proliferation by inhibiting PRV gene expression.

### 3.4. Luteolin Inhibits the Expression of the PRV gB Protein

UL27 encodes the gB protein, which plays a role in viral entry, fusion, and intercellular diffusion. The above experiments proved that luteolin can inhibit the expression of UL27 at the gene level. To study whether luteolin affects the presentation of the UL27 gene at the protein level, we first simulated and calculated the binding possibility between luteolin and the gB protein by molecular docking technology and predicted the possible binding sites between crystal gB and luteolin by AutoDock. The binding energy was calculated by molecular docking, as described in Section 2. The optimal calculation results showed that luteolin could bind to the gB protein, and the binding energy was −5.51 kcal/mol. Luteolin can form hydrogen bonds with gB amino acids GLU-141 and ALA-584, which may affect the activity of the gB protein (Figure 5A). To further verify the reliability of the molecular docking results, we conducted a Western blot experiment to ascertain the effect of luteolin on the expression level of gB protein. The results showed that the expression level of gB protein in infected PRV cells decreased in each treatment group compared with the virus control group (Figure 5B).

### 3.5. Luteolin Decreased the Apoptosis Rate of PRV-Infected Cells

The annexin V-FITC/PI flow cytometry results are shown in Figure 6. In PK15 cells, the apoptosis rates of the normal cell group, virus control group, and 70 µM luteolin treatment group were 9.1%, 18.5%, and 10.4%, respectively. The results showed that PRV could induce PK15 cell apoptosis, and 70 µM luteolin could inhibit PRV-induced apoptosis within 48 h. These results suggested that luteolin can inhibit the apoptosis of infected PRV cells and inhibit viral proliferation.

### 3.6. Luteolin Inhibits PRV Infection In Vivo

We studied the antiviral effect of luteolin in vivo. Five-week-old female mice weighing 18–20g were randomly divided into three groups with six mice in each group. The specific grouping and processing methods are shown in Table 2. The mice were kept in the experimental animal center for one week, with a normal diet and free access to drinking water. The development and survival of all mice were normal. Group B mice had obvious neurological symptoms on day three, decreased appetite, and shortness of breath and died quickly after onset, and all mice died on day six. All mice in group C developed disease on dayfour, which delayed the disease onset time and improved the survival rate by 50%, as shown in Figure 7A. The weight change in the mice showed that compared with the weight growth of group A mice, that of group B mice was slower, and the weight fluctuation of group C mice was slightly larger, which can promote the weight growth of mice to resist the slow weight growth caused by the viral infection, as shown in Figure 7B.

To determine the protective effect of luteolin on organs in PRV-infected mice, we collected and weighed the liver, kidney, heart, lung, and brain of the experimental mice in each group and extracted and quantified the viral DNA. The results are shown in Figure 8A. By comparing the viral content of each organ between the virus group and the luteolin treatment group, we found that the PRV DNA copies of the luteolin treatment group were significantly lower than those of the PRV group. Luteolin substantially reduced the viral content in the mouse liver, kidney, heart, lung, and brain. To further study the protective effect of luteolin on brain injury caused by PRV, we collected three groups of brain tissue sections and performed a histopathological examination, as shown in Figure 8B–D. There were sieve reticular softening foci and neuronal degeneration and necrosis in the brains of mice in the virus group, a small amount of lymphocyte infiltration in the luteolin group, and no specific symptoms in the brains of mice in the normal group. As expected, luteolin reduced PRV infection in an in vivo model. The study was based on experimental animal welfare and ethical principles. The design scheme was optimized, and the number of animals needed was strictly planned; thus, 18 BALB/c mice were used as planned. After the experiment, the mice were euthanized and ultimately incinerated.

### 3.7. Detection of Serum Cytokine Levels

Cytokines participate in immune and inflammatory responses and play a key role in protecting the body from foreign pathogens. Therefore, we measured the levels of TNF-α, IFN-γ, and IL-4 in mice. The results are shown in Figure 9. TNF-α and IFN-γ, which are proinflammatory cytokines, promote acute inflammation in the resistance response to infection. PRV infection caused the IFN-γ, TNF-α, and IL-4 levels to increase. Compared with those in the virus group, the levels of IFN-γ in the luteolin treatment group were elevated, and the levels of IL-4 and TNF-α were decreased. The results indicated that luteolin upregulated IFN-γ expression and downregulated IL-4 and TNF-α expression in the resistance response to PRV infection.

### 3.8. Determination of the Oxidative Stress Index

PRV infection can influence an animal’s oxidative stress response by promoting a reduction in the activity of antioxidant enzymes and an increase in the release of reactive oxygen species (ROS). In this study, we measured the concentrations of superoxide dismutase (SOD) and glutathione (GSH) in mouse serum. Figure 10 shows the results. We found that compared with those in normal mice, the levels of SOD and GSH in the serum of virus-infected mice decreased, and the levels of SOD and GSH in luteolin-treated mice increased significantly. The results showed that PRV infection caused oxidative damage. Luteolin could upregulate the expression of SOD and GSH to inhibit the oxidative stress response caused by PRV infection.

## 4. Discussion

Pseudorabies disease (PR) is a highly contagious disease caused by PRV. Once PRV infection occurs, it is difficult to control in pigs. Since 2011, the emergence of PRV mutant strains has increased the difficulty of PR prevention and control in China [27]. In addition, studies have shown that PRV has the risk of infecting people and can lead to encephalitis [5]. PRV can infect many animals [28] and causes severe neurological symptoms after infection. The genomic structure of PRV is complex, and the virus can produce obvious cytopathy when cultured in cells. The virus has always been a vital model virus for virologists to study the universality of viral hosts and nervous system damage [29]. Therefore, developing new insights into the development of antiviral methods is important.

Luteolin is a natural flavonoid found in bryophytes, ferns, pines, and magnolias [30]. At present, the antiviral effect of luteolin has been widely studied. Some studies have reported the inhibitory effect of flavonoids in extracts containing luteolin on the H1N1 influenza virus [31]. Lv demonstrated that luteolin inhibited EV71 gene expression and RNA synthesis to inhibit EV71 replication and can inhibit EV71-induced apoptosis and ROS production [32]. Krishnan showed that the ethanol extract of Cynodon dactylon containing luteolin inhibited the synthesis of viral mRNA in vitro demonstrating an anti-chikungunya virus effect [23]. However, few studies have investigated the antiviral properties and mechanism of luteolin against PRV. In this study, we studied the anti-PRV activity of luteolin in vivo and in vitro and discussed the primary mechanism of luteolin against PRV for the first time. First, we found that luteolin had no significant effect on cell activity at a concentration of 70 μM in vitro, and its CC_50_ was 148.1 μM ± 1.15. According to the cytotoxicity results, luteolin had antiviral activity against the PRV-TJ strain at noncytotoxic concentrations, and luteolin inhibited PRV activity in a concentration-dependent manner, with a selection index of 5.64 ± 0.04.

The replication cycle of PRV mainly includes adsorption, entry, replication, assembly, and viral particle release. Moreover, each stage can represent a key and indispensable target of antiviral drugs [2]. Massimo Ritt et al. found that luteolin inhibited the HSV-2 replication cycle [33]. Lin showed that luteolin targets the postattachment stage of EV71 and CA16 infection by inhibiting viral RNA replication [34]. Wu suggested that luteolin inhibited EBV reactivation by inhibiting the promoter activity of ZP and RP [35]. This study investigated the effects of luteolin on viral adsorption, entry, and replication through three modes of action. We found that luteolin had different degrees of influence. These three stages had the most apparent inhibitory effect on the replication stage (Figure 3). From this point in time, luteolin could inhibit the immediate early gene of the virus (IE180) within 72 h. The expression of early genes (EP0 and UL54) and other genes (UL9, UL27, and UL29) (Figure 4), also showed that luteolin could inhibit the replication of viral RNA by affecting the replication stage of PRV infection. The interaction between antiviral particles and viral proteins is also a way to exert antiviral activity. Yan found that luteolin targeting β-Cop expression inhibited influenza virus replication and reduced the production of influenza A virus in vitro [24]. Sirin found that luteolin exerted anti-FMDV activity by reacting with key enzyme residues of 3Cpro [36]. The PRV gB protein is a membrane protein encoded by the UL27 gene. It is involved in the attachment of viral particles to the surface of host cells, the fusion of the viral envelope and plasma membrane, and the intercellular transmission of viral particles. This study used AutoDock to predict the interaction between the PRV gB protein and luteolin, and molecular docking was used to calculate the binding energy. The results showed that luteolin could bind to the gB protein. In Figure 5A, the molecular binding model of gB luteolin shows that luteolin can form hydrogen bonds with gB amino acids GLU-141 and ALA-584, which may affect the activity of the gB protein. The WB results also showed that luteolin inhibited the expression of the gB protein, which indicated that luteolin reduced the level of the PRV gB protein to reduce the attachment of viral particles, membrane fusion, and intercellular transmission. The anti-apoptotic effect of luteolin is well known. For example, luteolin has been shown to induce cancer cell apoptosis by inhibiting the dual inhibition of the MAPK and PI3K signaling pathways in gastric cancer cells [37], promote neuronal survival and reduce neuronal apoptosis through the phosphatidylinositol 3-kinase/Akt signaling pathway [38], and attenuate methamphetamine-induced apoptosis by inhibiting the PI3K/Akt pathway [39]. In our study, luteolin reduced PRV-induced apoptosis to promote the survival of infected cells and inhibit viral proliferation.

Studies have shown that luteolin improves the survival rate of newborn mice challenged with a lethal dose of EV71 [40], and luteolin reduces the viremia of mice to exert anti-DENV activity [22]. In this study, the anti-PRV evaluation of luteolin in vivo found that luteolin can delay the disease onset time in mice infected with PRV, improve the survival rate of infected mice, reduce the viral load in mouse organs and reduce the damage of the virus to brain tissue, which is consistent with Li et al. [41]. After viral infection, the body protects cells from viral infection by regulating the expression of cytokines, such as interferon, interleukin, and tumor necrosis factor, through nonspecific immunity. IFN-γ can activate macrophages and promote the production of the inflammatory factors IL-12, TNF-α, and IL-1β to regulate the body’s immunity [42]. IFN-γ-induced NO production inhibits viral replication, dilates blood vessels, and reduces blood flow to allow immune cells to extravasate to infected and inflammatory sites [43]. TNF-α is an inflammatory cytokine and can cooperate with IFN to play an antiviral role. Research shows that TNF-α and IFN-γ synergistically inhibit the expression and replication of the hepatitis B virus (HBV) gene [44], and TNF-α overexpression can exacerbate the body’s inflammatory response. IL-4 is a cytokine produced by T cells, mast cells, basophils, and eosinophils. It plays a central role in the body’s immune response, promoting the development of the Th2 response and inhibiting the Th1 response [45]. In this study, we measured the levels of cytokines in the serum of animals infected with PRV. The results showed that luteolin upregulated the expression of IFN-γ, downregulated the expression of TNF-α and IL-4, inhibited the inflammatory response caused by a viral infection, and improved the expression of interferon. We speculated that luteolin treatment might inhibit Th2 cell-mediated cellular immunity and promote Th1 cell-mediated humoral immunity and IFN-γ antiviral function. PRV infection leads to an oxidative stress response. Studies have shown that oxidative stress and free radicals caused by PRV infection can lead to DNA damage and apoptosis [46]. The levels of SOD and GSH in PRV-infected mice decreased, and luteolin treatment increased these levels. We speculated that luteolin might reduce DNA damage by weakening the oxidation reaction caused by PRV and then inhibiting the apoptosis induced by PRV infection. However, many factors affect the antiviral activity of luteolin in vivo. Whether luteolin can reduce PRV infection in other animals needs further study.

## 5. Conclusions

This study showed that luteolin had significant anti-PRV activity in vivo and in vitro. Luteolin effectively inhibited PRV replication by inhibiting the expression of viral mRNA and gB protein. Luteolin also inhibited the apoptosis of infected cells. Animal experiments showed that luteolin delayed the disease onset time in mice, improved the survival rate, inhibited PRV replication in mouse organs, and alleviated inflammation and oxidative response in infected mice. In conclusion, luteolin can be used as a therapeutic chemical and has a robust protective effect against PRV infection in vitro and in vivo.

## Figures and Tables

**Figure 1 animals-13-00761-f001:**
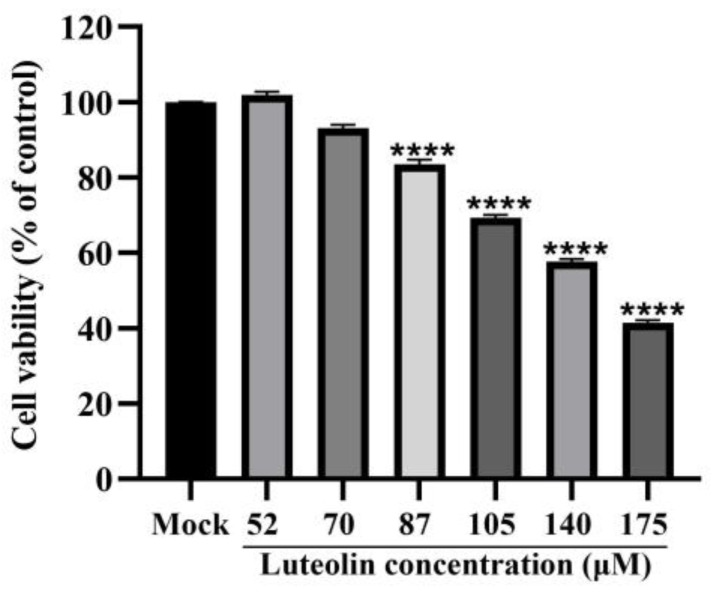
CCK8 assay was used to determine the toxicity of luteolin toward PK15 cells compared with normal cells at 52, 70, 87, 105, 140, and 175 μM (**** *p* < 0.0001).

**Figure 2 animals-13-00761-f002:**
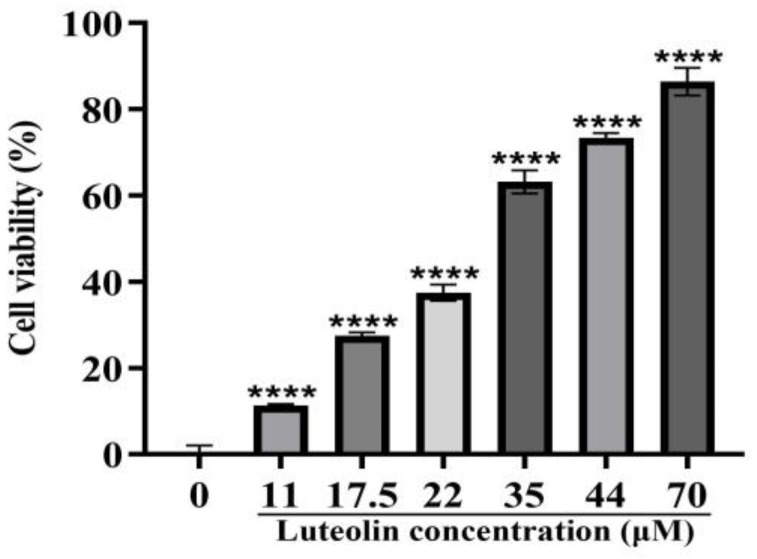
CCK8 assay was used to determine the antiviral activity of luteolin at 0, 11, 17.5, 22, 35, 44, and 70 μM. Different concentrations of luteolin were added 1 h after PRV infection. The cell survival rate after 48 h of continuous culture was determined **** *p* < 0.0001).

**Figure 3 animals-13-00761-f003:**
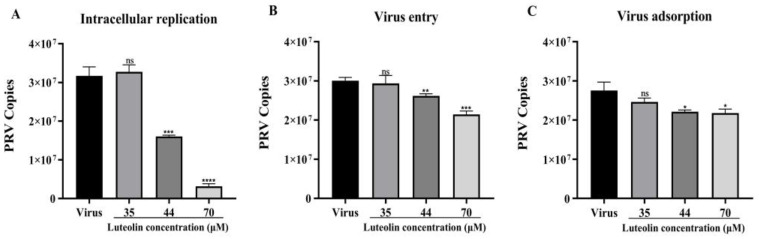
FQ-PCR was used to detect the effect of luteolin on the copy number of three stages of viral infection. (**A**) Luteolin was added at the viral replication stage. (**B**) Luteolin was added at the viral entry stage. (**C**) Luteolin was added at the viral adsorption stage. (ns: nonsignificant,* *p* < 0.05, ** *p* < 0.01, *** *p* < 0.001 and **** *p* < 0.0001).

**Figure 4 animals-13-00761-f004:**
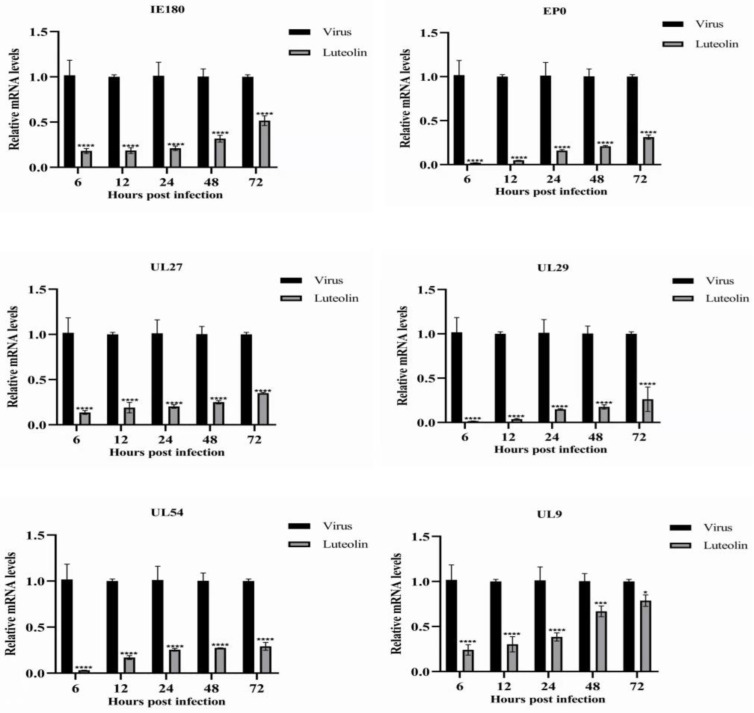
Luteolin inhibited the expression of a PRV gene. Gene expression levels of PRV in the presence or absence of luteolin (70 μM) were assayed at 6, 12, 24, 48, and 72 hpi (n = 3, in each group). (* *p* < 0.05, *** *p* < 0.001 and **** *p* < 0.0001).

**Figure 5 animals-13-00761-f005:**
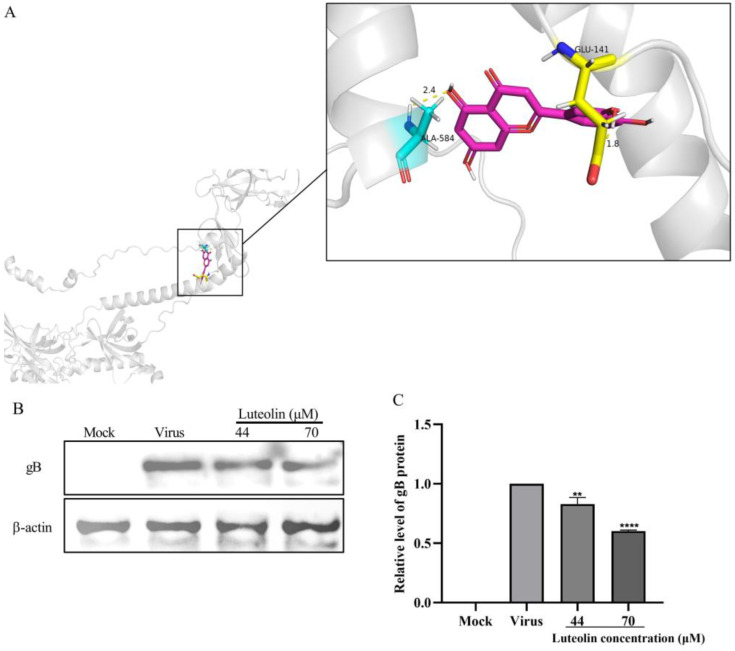
Luteolin inhibited the expression of the PRV gB protein. (**A**) Optimal binding results of the PRV gB protein to luteolin. The pink rod structure represents luteolin, and the PRV gB complex protein is represented by the shadow. The contact amino acid residues of the gB protein are displayed as rods and labeled. Two hydrogen bonds (yellow) were observed between luteolin and the gB amino acid residues. (**B**) After PRV infection for 1 h, luteolin was added, and the cells were cultured for 48 h. Then, the viral gB protein was measured by Western blotting. (**C**) Quantification of the gB protein (** *p* < 0.01, **** *p* < 0.0001).

**Figure 6 animals-13-00761-f006:**
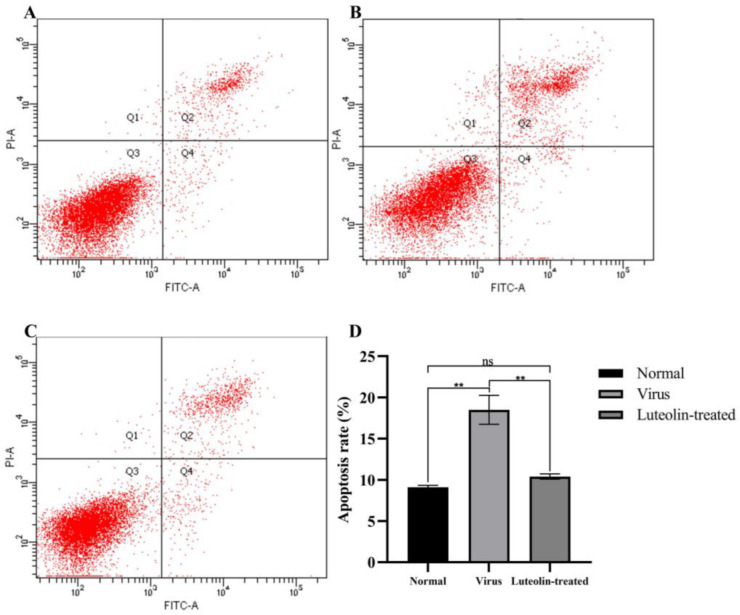
Luteolin inhibited the apoptosis of infected PRV cells at 48 h. (**A**) Normal cell group, (**B**) PRV-infected group, (**C**) 70 µM luteolin treatment group after viral infection, and (**D**) quantitative evaluation of the cell apoptosis rate. The data are expressed as the mean ± standard deviation (ns: nonsignificant, ** *p* < 0.01, compared with the normal cell group).

**Figure 7 animals-13-00761-f007:**
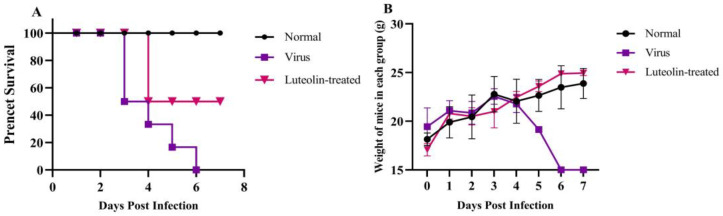
Changes in the survival rate and body weight of mice in each group. (**A**) Survival rate of mice in each group. (**B**) Changes in the body weight of mice in each group.

**Figure 8 animals-13-00761-f008:**
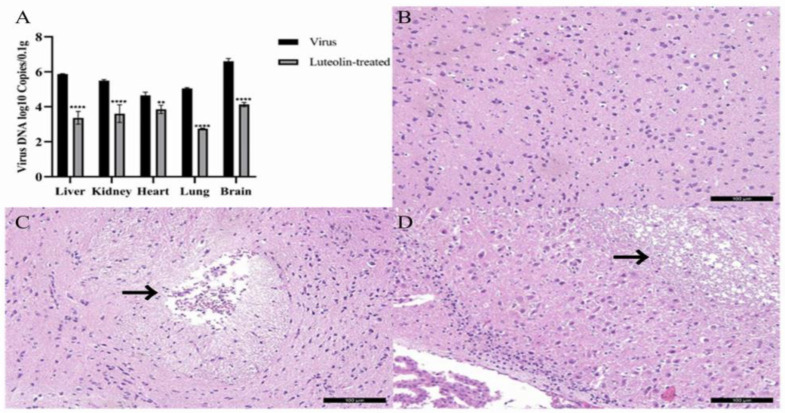
Virus copy number in different organs of mice and three groups of mouse brain tissue sections. (**A**) Virus copy number in different mouse organs (** *p* < 0.01, and **** *p* < 0.0001). (**B**) Normal group mouse brain tissue sections. (**C**) Virus group mouse brain tissue sections. The site shown by the arrow is the lesion site. (**D**) Luteolin-treated group mouse brain tissue sections. The site shown by the arrow is the lesion site.

**Figure 9 animals-13-00761-f009:**
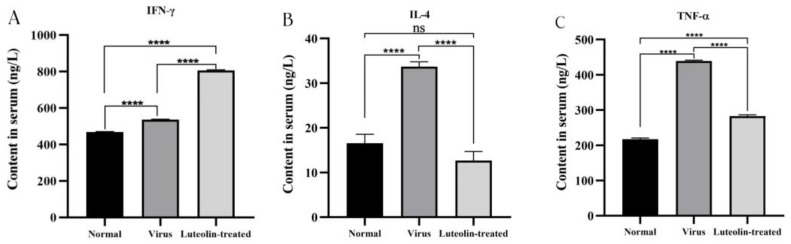
Concentrations of TNF-α, IL-4, and IFN-γ in the serum of mice in each group, (**A**) IFN-γ, (**B**) IL-4, (**C**) TNF-α (ns: nonsignificant, **** *p* < 0.0001).

**Figure 10 animals-13-00761-f010:**
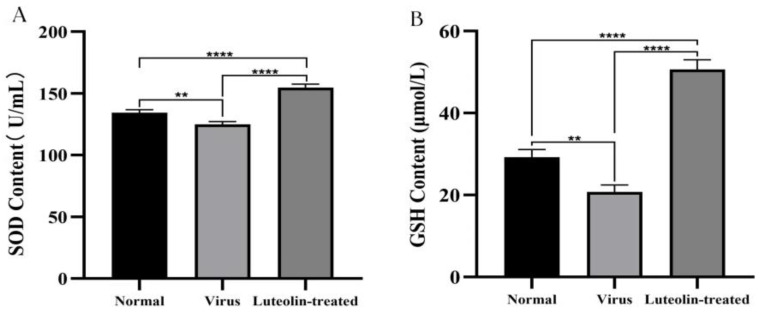
Levels of SOD and GSH in the serum of mice in each group (**A**) SOD, and (**B**) GSH (** *p* < 0.01, and **** *p* < 0.0001).

**Table 1 animals-13-00761-t001:** Primers and probe for real-time PCR for PRV detection.

Name	Sequence (5′-3′)
IE180-F	CGCTCCACCAACAACC
IE180-R	TCGTCCTCGTCCCAGA
EP0-F	GGGCGTGGGTGTTT
EP0-R	GCTTTATGGGCAGGT
UL9-F	CTGGCGAGACGAATGG
UL9-R	TGGTGGGCGAGTAGAGC
UL27-F	CGGTCACCTTGTGGTTGTTG
UL27-R	GGATCGCCGTGCTCTTCA
UL29-F	CCCCGTGAGGCTGTTGA
UL29-R	GCGCTTCTCGGTGGACTA
UL54-F	GGGTGTAGGTGACGATGC
UL54-R	AGATGGTGCTCCTGAACG
PRV Probe	HEX-CCGCGTGCACCACGAAGCCT-BHQ1
Β-actin-F	TGCGGGACATCAAGGAGAA
Β-actin-R	AGGAAGGAGGGCTGGAAGA

**Table 2 animals-13-00761-t002:** Grouping of in vivo experiments.

Numbering	Group	Intraplantar
A	Normal	100 μL DMEM
B	Virus	100 μL PRV-TJ
C	Luteolin-treated	100 μL PRV-TJ + 100 mg/kg Luteolin

**Table 3 animals-13-00761-t003:** Inhibitory effects of luteolin on PRV in vitro.

Compound	CC_50_ (μM)	IC_50_ (μM)	SI
Luteolin	148.1 ± 1.15	26.24 ± 0.01	5.64 ± 0.04

## Data Availability

The data presented in this study are available in the article.

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
