# Peer review of "Antiviral Activity of Luteolin against Pseudorabies Virus In Vitro and In Vivo"

_animals, 2023, doi:10.3390/ani13040761_

Round 1

Reviewer 1 Report

The work from Men et al is an original study that aims to investigate the protective effect of Luteolin against Pseudorabies virus infection.

Whereas the subject is original, I think that some major issues should be solved before publication in Viruses.

1. Fig 3: Experimental procedures for viral adsorption, invasion, and replication are missing.

2. Fig 3: The antiviral of Luteolin should be verified by different methods. 

3. The dose of PRV was 100TCID50? It's too low. When the dose of PRV is high, whether Luteolin will have an antiviral effect.

4. The decrease in gB was related to the antiviral effect of luteolin, which was not related to molecular docking.A nd the fact that luteolin affects gB activity is contradicted by the fact that it reduces gB activity.

5. Luteolin decreased apoptosis, but the relationship between apoptosis and PRV infection was not elucidated.

6. The results indicated that luteolin upregulated IFN-γ expression and downregulated IL-4 and TNF-α expression in the resistance response to PRV infection. The conclusion was not valid.

7. What is the role of ROS in luteolin antiviral activity?

Author Response

请参阅附件。

Reviewer 2 Report

In the current manuscript, authors have studied the antiviral property of Luteolin against the pseudorabies virus. 

Below is my concern with this manuscript:

  1. Include appropriate references in the introduction. Several statements were made without citing references.
  2. Why only choose PK15 and no nasal epithelium cell line for this study?
  3. Line 87: Add volume (uL) of the medium in which respective concentration of Luteolin was used to treat cells.
  4. No forward primer for IE180?
  5. Table.2: What is the route of inoculations? Intraplantar or SC or IM?
  6. Fig.8. Show an arrow that can help the reader to understand what lesions you attempt to show here.
  7. It would be great if you include immunofluorescence images for virus replication in addition to qPCR.
  8. Mechanism of Luteolin mediated suppression of PRV replication is remains unknown.
  9. Material and methods section 2.4 needs to explain in more detail.

Author Response

请参阅附件

Round 2

Reviewer 1 Report

IFA or wb needs to check the antivirus.